# Covalent Adaptable Networks from Polyacrylates Based on Oxime–Urethane Bond Exchange Reaction

**DOI:** 10.3390/ijms252312897

**Published:** 2024-11-30

**Authors:** Yu Sotoyama, Naoto Iwata, Seiichi Furumi

**Affiliations:** Department of Chemistry, Graduate School of Science, Tokyo University of Science, 1-3 Kagurazaka, Shinjuku, Tokyo 162-8601, Japan; 1323564@ed.tus.ac.jp (Y.S.); n-iwata@rs.tus.ac.jp (N.I.)

**Keywords:** polyacrylate, covalent adaptable network, oxime–urethane bond exchange reaction, miscibility

## Abstract

Covalent adaptable networks (CANs) are polymer networks cross-linked via dynamic covalent bonds that can proceed with bond exchange reactions upon applying external stimuli. In this report, a series of cross-linked polyacrylate films were fabricated by changing the combination of acrylate monomer and the amount of diacrylate cross-linker possessing oxime–urethane bonds as a kind of dynamic covalent bond to evaluate their rheological relaxation properties. Model analysis of the experimental relaxation curves of cross-linked polyacrylate films was conducted by assuming that they consist of two types of relaxation, one of which is related to the oxime–urethane bond exchange reaction, and another of which is associated with the melting of the aggregated cross-linker. It was found that the contribution from the relaxation due to the bond exchange reaction becomes dominant only when the normal-alkyl acrylates are used as a monomer. The relaxation time was almost constant even when the amount of the cross-linker was adjusted. Moreover, it was also indicated that the miscibility of the cross-linker is very important for the fabrication of CANs with good self-healing ability and reprocessability.

## 1. Introduction

Polyacrylates are one of the most important polymer materials in our daily life. For instance, the global production capacity of butyl acrylate is 3.45 million tons/year [1] as it can be used in adhesives [2], sealants [3], and so forth. Moreover, they are the main component of acrylic rubbers when they are copolymerized with other monomers or cross-linkers [4,5,6,7].

However, nowadays we encounter serious problems in both resource and environmental perspectives. Since acrylates are generally synthesized from petroleum resources, it is important to improve its recyclability considering the recent growing concerns about environmental issues [8,9]. Conventional polymers can be categorized into thermoplastics or thermosets. Thermoplastics can be reprocessed arising from their flowability at heating treatment, but they are inevitably dissolved when soaked in good solvents or are easily melted by heating. On the other hand, thermosets are advantageous because they hardly dissolve in solvents and flow upon heating, but they cannot be reprocessed because of the robust cross-linked structure of the polymer networks in the three-dimensional microscopic scale.

Covalent adaptable networks (CANs) are the polymer networks cross-linked via dynamic covalent bonds that can proceed bond exchange reactions upon applying external stimuli [10,11,12,13]. CANs exhibit intrinsic physical properties such as solvent resistance and thermal stability in the absence of external stimuli, but they can be reprocessed like thermoplastics by applying external stimuli arising from the activation of bond exchange reactions between dynamic covalent bonds. Previously, CANs utilizing transesterification have been widely studied because the bond exchange reaction can be easily proceeded by the activation of transesterification [11,14,15,16]. Such CANs can be easily prepared by the molecular design of polymer networks having pendant hydroxy groups and a cross-linked structure via ester bonds. This strategy cannot apply to the fabrication of polyacrylate-based CANs because ester moieties of polyacrylates may also contribute to the bond exchange reaction triggered by transesterification. In addition, polymerization of acrylates with functional groups is generally difficult because protection of functional groups becomes necessary to prevent undesired side reactions during the polymerization process. Moreover, polymerization may not proceed uniformly because of differences in the reactivity ratios when copolymerized with other acrylate monomers.

Due to the obstacles mentioned, the number of reports on CANs from polyacrylates is limited. One of the promising strategies to introduce dynamic covalent bonds into polyacrylate networks is a copolymerization with a cross-linker possessing dynamic covalent bonds. The versatility of this strategy is beneficial as it can be applied to many acrylate monomers [17,18]. For example, Li and colleagues succeeded in fabricating polymethacrylate-based CANs by cross-linking methacrylate monomers via a dimethacrylate cross-linker containing boronic ester bonds [17].

In this study, we adopted oxime–urethane bonds as a type of dynamic covalent bond since the oxime–urethane bond exchange reaction progresses independently regardless of the presence of ester bonds in polyacrylates. In previous studies, polyurethane-based thermoset materials and elastomers incorporating oxime–urethane bonds as dynamic covalent bonds have been fabricated [19,20,21,22,23,24]. Moreover, the oxime–urethane bond exchange reaction proceeds in the absence of a catalyst while preserving ester moieties of polyacrylates. This is very beneficial from an experimental perspective because CANs utilizing transesterification reactions require a base catalyst, which must have good miscibility to the polymer network [25]. By copolymerization of acrylate monomers with a diacrylate cross-linker with an oxime–urethane bond (DOUDA), a new class of CANs based on polyacrylates were successfully prepared. Moreover, we investigated the reprocessability of a series of cross-linked polyacrylate films fabricated by changing the combination of acrylate monomers and the amount of cross-linkers. It was indicated that the miscibility of DOUDA is very important for the fabrication of CANs that show good reprocessability because aggregated DOUDA hindered the bond exchange reaction.

## 2. Results and Discussion

### 2.1. Synthesis of Cross-Linked Films

The synthesis and characterization of DOUDA was adopted from previous reports [26,27]. Cross-linked films were prepared by bulk photopolymerization at 110 °C of a mixture of DOUDA, which is a diacrylate containing oxime–urethane bonds (Figure 1A), an acrylate monomer, and 2-hydroxy-2-methylpropiophenone (HMPP) as a photo radical initiator. In this study, butyl acrylate (BA, Figure 1B), octyl acrylate (OA, Figure 1C), 2-methoxyethyl acrylate (MEA, Figure 1D), and 2-(2-ethoxyethoxy)ethyl acrylate (EEA, Figure 1E) were used. The chemical structure of the cross-linked film, in which the polyacrylate main chain is cross-linked via DOUDA, is presented in Figure 1F. The fabrication conditions of cross-linked films are listed in Table 1. Hereafter, samples are coded as P*x*-*y*, where *x* is the abbreviation of the acrylate monomer and *y* is the mole fraction of DOUDA. The cross-linked films were characterized by Fourier transform infrared (FT-IR) spectroscopy. As an example, FT-IR spectra of BA and PBA-0.5 measured by the attenuated total reflection (ATR) method are shown in Figure 2. In the FT-IR spectrum of BA, two peaks at ~1620 and ~1637 cm^−1^ appeared, arising from the C=C stretching vibration. However, these peaks were not observed for the FT-IR spectrum of PBA-0.5 of the UV-irradiated side. Concomitantly, a peak originating from the C=O stretching vibration shifted from 1725 cm^−1^ to 1735 cm^−1^ after photopolymerization with DOUDA. Furthermore, a decrease in transmittance originating from the N-O bond was observed at ~927 cm^−1^ after copolymerization with DOUDA (Figure 2) [28]. These results indicate that the oxime–urethane bonds are incorporated into the cross-linked film. When the film of PBA-0.5 was immersed in tetrahydrofuran (THF), PBA-0.5 interestingly swelled up to 745% by weight without dissolving in THF (Table 1, 4th column). From these results, it is plausible that the complete conversion of BA and DOUDA as well as cross-linking of PBA via DOUDA proceed. Moreover, the ATR FT-IR spectra of PBA-0.5 were almost identical when measured on both sides of the film (Appendix A), suggesting that the polymerization of BA and DOUDA homogeneously proceeds in the film thickness direction. Such disappearance of the peaks assigned to the C=C stretching vibration and the shift of a peak from C=O stretching vibration was also observed for all of the cross-linked films regardless of the difference in the molar ratio of DOUDA or the chemical structure of the acrylate monomers. From these results, it was clear that all acrylates are fully cross-linked via DOUDA and their cross-linked densities can be controlled by changing the molar ratio of DOUDA to acrylate monomers.

The rheological properties of the cross-linked films were measured using a rheometer. The glass transition temperature (*T*_g_) of the cross-linked films was found to be almost the same as that of the homopolymer regardless of the molar ratio of the DOUDA (Table 1, 5th column) [29,30]. It should be noted that the molar ratio of DOUDA was low in all cross-linked films. When utilizing BA or OA, the storage modulus (*G*’) at 30 °C almost proportionally increased with an enlargement of the molar ratio of the DOUDA (Table 1, 6th column). This is reasonable since the cross-link density can be controlled by the molar ratio of DOUDA, owing to the complete conversion of acrylate monomers and DOUDA.

### 2.2. Stress Relaxation Measurements

The results of the stress relaxation measurements of PBA-0.5 are shown in Figure 3 and Appendix A. A master curve was constructed using the time–temperature superposition principle at a reference temperature of 150 °C [31]. The horizonal (*a*_T_) and vertical (*b*_T_) shift factors used for the construction of the master curve are listed in Appendix A. Each stress relaxation curve measured at the temperature between 150 °C and 180 °C was overlapped to be a smooth curve on a time scale of ~0–10^2^ s. In contrast, the overlap of the stress relaxation curves became worse on a time scale of ~10^2^–10^4^ s (Figure 3A). This result indicates that another relaxation with different activation energy is occurring on a time scale of ~10^2^–10^4^ s. Therefore, a model analysis of the experimental relaxation curve was performed assuming that the stress relaxation curve of PBA-0.5 consisted of two types of relaxation, one of which is related to the oxime–urethane bond exchange reaction. According to the previous study by Hayashi and colleagues, the relaxation modulus *G*(*t*) with initial modulus *G*_0_ can be numerically expressed by the quadratic Kohlrausch–Williams–Watts (KWW) type function as follows [32]:(1)$$\frac{G\left(t\right)}{G_{0}}=A_{1}exp\left\{-{\left(\frac{t}{\tau_{1}}\right)}^{\beta_{1}}\right\}+A_{2}exp\left\{-{\left(\frac{t}{\tau_{2}}\right)}^{\beta_{2}}\right\}$$
where *τ*_1_ and *τ*_2_ represent the relaxation times that correspond to the bond exchange reaction, the other relaxation, *β*_1_ and *β*_2_, are the distribution of relaxation time, and *A*_1_ and *A*_2_ are the contributions of two relaxations. In this study, *τ*_1_ and *τ*_2_ were defined in ascending order of relaxation time. The experimental relaxation curve was fitted well by adopting Equation (1), as shown in Figure 3B. All fitting parameters are listed in Appendix A. In the case of relaxation with shorter time scale, the average relaxation time <*τ*_1_> considering the distribution of relaxation time can be calculated from Equation (2), where Γ is the gamma function.
(2)$$\langle \tau_{1}\rangle =\int_{0}^{\infty }exp\left\{-{\left(\frac{t}{\tau_{1}}\right)}^{\beta }\right\}dt=\frac{\tau_{1}\;\Gamma \left(\frac{1}{\beta }\right)}{\beta }$$

The Arrhenius-type relationship was observed for the calculated average relaxation time <*τ*_1_>, with activation energy (*E*_a,1_) being 158 kJ/mol (Table 1, Equation (3)) as follows:(3)$$\tau_{1}=\tau_{0,1}exp\left(\frac{E_{a,1}}{RT}\right)$$
where *τ*_0,1_ stands for the pre-exponential factor, *R* is the gas constant, and *T* is the absolute temperature. In the same way as <*τ*_1_>, when <*τ*_2_> was calculated for relaxation with a longer time scale, the Arrhenius-type relationship was also observed, thereby estimating the activation energy (*E*_a,2_) of 273 kJ/mol (Appendix A). Since the activation energy of oxime–urethane bond exchange reactions has been reported as 100–180 kJ/mol, according to the previous study by You and colleagues [26], it was suggested that the relaxation with a shorter relaxation time originated from the oxime–urethane bond exchange reaction. Moreover, the activation energy calculated from *a*_T_ of the master curve (*E*_a,sf_) was 153 kJ/mol, which was in agreement with *E*_a,1_ (Table 1). Since the relaxation curves for ~10^2^–10^4^ s at each temperature shifted to the longer time side as the temperature decreased, it is reasonable that *E*_a,2_ becomes larger than *E*_a,1_ arising from the relatively large temperature dependency of the relaxation behavior. Based on these facts, it is strongly reasonable to construct the fitting curve using the quadratic KWW type function assuming two types of relaxation as shown in Equation (1). This tendency was also observed for other cross-linked films utilizing BA or OA as an acrylate monomer. The results of stress relaxation measurements for PBA-1.0, PBA-2.0, POA-0.5, POA-1.0, and POA-2.0 are also shown in Appendix A. In all cases, the stress relaxation curves were well-fitted with Equation (1), suggesting the existence of two different relaxation modes. The proposed mechanism for the emergence of longer time-scale relaxation will be described later.

The long-timescale stress relaxation might occur due to the melting of the aggregated DOUDA cross-linker as it appeared for all cross-linked films utilizing either BA or OA regardless of the difference in the molar ratio DOUDA. In fact, DOUDA could not dissolve in BA or OA at room temperature because of its low miscibility. Although DOUDA was mixed with acrylate monomers by heating above its melting point to fabricate the cross-linked films, DOUDA might not be completely miscible with alkyl acrylate monomers even in the liquid state.

The relatively low miscibility of DOUDA with acrylate monomers was especially apparent when MEA or EEA was used. At first, we assumed that acrylate monomers possessing oxygen atoms in the side chain might improve the miscibility with DOUDA because DOUDA is easily dissolved in highly polar organic solvents such as *N*,*N*-dimethylformamide or dimethyl sulfoxide. The results of the stress relaxation measurements of PMEA-0.5 are shown in Appendix A. The relaxation curves at 170 °C and 180 °C showed a slower decrease in *G*(*t*) at ~10^2^–10^4^ s, arising from the longer time scale stress relaxation by the melting of aggregated DOUDA. As the cross-linked film was fabricated with PBA-0.5, the experimental relaxation curves were fitted well by adopting Equation (1), as shown in Appendix A. However, the Arrhenius plot could not be constructed because the relaxation time became much longer than that of PBA-0.5 compared with that at the same temperature. In fact, the average value in long-timescale stress relaxation time of PMEA-0.5 at 180 °C was ~10 times longer than PBA-0.5 (Appendix A 7th column, Appendix A 7th column). The significantly longer relaxation time of PMEA-0.5 happened from the aggregated DOUDA because the miscibility between MEA and DOUDA was lower than that of BA and OA. This is evident from the enlargement of the contribution of longer time scale relaxation (*A*_2_ in Equation (1)) by the melting of aggregated DOUDA (Appendix A 11th column, Appendix A 11th column, Appendix A 11th column). Moreover, this tendency was also observed for PEEA-0.5 as its relaxation curve at 180 °C became extremely longer than that of PMEA-0.5 (Appendix A). From these results, it was suggested that the compatibility of DOUDA with MEA or EEA is much lower than that with BA or OA. This hypothesis can be supported by the peculiar rheological behaviors of cross-linked films. Appendix A shows the temperature dependence of the storage modulus (*G*’) and loss modulus (*G*”) of PBA-0.5 and PMEA-0.5. The characteristic peaks were not observed for PBA-0.5. In the case of PMEA-0.5, a peak in *G*’ and *G*” were observed at ~0 °C and above ~100 °C, respectively. An increase in *G*’ value at ~0 °C might be related to the aggregation of DOUDA and an increase in *G*” value is the melting of aggregated DOUDA (Appendix A, arrows). From these results, PMEA-0.5 and PEEA-0.5 were found to be difficult to apply as CANs. This is because the stress relaxation due to the bond exchange reaction is not dominant, arising from the low compatibility with DOUDA. In the case of conventional cross-linked polymers, the applied stress does not relax completely due to the formation of a three-dimensional polymer network structure by static bonds. On the other hand, stress relaxation was observed for all films because these are cross-linked by DOUDA. Thus, DOUDA has the potential to add self-healing abilities to cross-linked polymers as it enables the bond exchange reaction. However, it should be emphasized that the bond exchange reaction is greatly affected by the miscibility of DOUDA. Normal-alkyl acrylates such as BA and OA were found to be suitable for cross-linking with DOUDA. These results strongly suggest that it is important to incorporate dynamic covalent bonds uniformly into the networks of cross-linked polymers. However, it should be noted that the mole fraction of DOUDA must be less than 200:4, even for monomers with relatively good miscibility such as BA and OA. In fact, when POA-4.0 and POA-8.0 cross-linked films with a 200:8 or 200:16 molar fraction of DOUDA were prepared, both films were whitely turbid. Although the transmittance of POA-0.5 was 90%, POA-4.0 and POA-8.0 were less than 10% (Appendix A). Such a significant decrease of transmitted light strongly suggests that the aggregated DOUDA causes the light scattering phenomenon.

The relaxation time was almost constant even when the amount of DOUDA cross-linker was changed and when BA or OA was used as an acrylate monomer (Appendix A). As the molar ratio of DOUDA increased, the cross-link density of the films increased. In the case of CANs, such an increase in cross-link density might give rise to the reduction of stress relaxation time by changing the pre-exponential factor of the Arrhenius plot, indicating that the bond exchange reaction will likely proceed. In contrast, an increase in cross-link density might also contribute to the enlargement of relaxation time as it hinders the polymer chain motions. In the case of the cross-linked films from BA or OA, it can be assumed that these opposite effects offset each other so that the relaxation time is not relevant to the amount of cross-linker in the films. It is important to note that this is likely due to the sufficiently low molar ratio of the cross-linker, which does not affect the mobility of polymer chains. The relaxation times due to bond exchange for POA-0.5, POA1.0, and POA-2.0 were maintained (Appendix A), while *G*’ value at 30 °C increased with an enlargement of the molar ratio of DOUDA, suggesting the increase in cross-link density (Table 1, 6th column). Figure 4 shows the stress–strain curves of POA-0.5, POA1.0, and POA-2.0 in the compression process with the stress increasing to 20%. The Young’s modulus was ~13 kPa regardless of the difference in the molar ratio of DOUDA. The hysteresis loss rate was ~24% for all films, which is comparable to that of commercially available chloroprene rubbers. This suggested that the bond exchange reaction is completely frozen at room temperature similar to the conventional thermosets. Young’s modulus remained constant upon repeated circles of compression, while the maximum point stress gradually decreased to 84% of the initial value (Appendix A). Therefore, we succeeded in fabricating CANs with different flexibility while maintaining the bond-exchange properties by changing the monomer and the molar ratio of DOUDA.

### 2.3. Reprocessability

We demonstrated herein the reprocessability of the cross-linked film by utilizing POA-0.5. POA-0.5 was chosen because of its shortest relaxation time at 180 °C. When broken pieces of POA-0.5 were hot-pressed at 200 °C for 10 min, they formed into a single sheet (Figure 5A). Interestingly, the cross-linked film of POA-0.5 exhibited high optical transparency even after reprocessing at 200 °C. When stress relaxation measurements were performed again, the stress relaxed in the same time scale when compared with as-prepared films. Fitting curves showed that the relaxation due to the bond exchange reaction occurred on the same time scale and with almost the same activation energy (Figure 5B,C, Appendix A). The FT-IR spectrum was unchanged before and after reprocessing, confirming that the chemical composition of POA-0.5 was maintained (Appendix A). Moreover, the *G*’ value at 30 °C was 17.8 kPa and was found to be unchanged before and after reprocessing. This is also evident from Young’s moduli determined from the linear regime of the stress–strain curves during compression were consistent before and after reprocessing. The increase in the maximum stress in the stress–strain curve is probably due to oxidation, which might be unavoidable as long as POA-0.5 is heated during reprocessing (Appendix A). Thus, it was demonstrated that the DOUDA can give polyacrylate reprocessability without changing the properties of the cross-linked film before and after reprocessing.

## 3. Materials and Methods

### 3.1. Materials

Dimethylglyoxime (98.0%), 1,8-diazabicyclo[5.4.0]-7-undecene (98.0%), 2-hydroxy-2-methylpropiophenone (HMPP, 96.0%), butyl acrylate (BA, 99.0%), and octyl acrylate (OA, 98.0%) were obtained from the Tokyo Chemical Industry Co., Ltd. (Tokyo, Japan). 2-Acryloyloxyethyl isocyanate (97.0%) was kindly donated from the Resonac Corporation (Tokyo, Japan). 2-Methoxyethyl acrylate (MEA) and 2-(2-ethoxyethoxy)ethyl acrylate (EEA) were kindly gifted from the Osaka Organic Chemical Industry Ltd. (Osaka, Japan). All reagents were used without further purification.

### 3.2. Synthesis of DOUDA Cross-Linker

DOUDA was synthesized according to the typical procedures reported by individual research groups of You and colleagues [26], as well as Zhang and colleagues [27]. In a 50 mL vial bottle with nitrogen substitution, 1.16 g of dimethylglyoxime (10.0 mmol) and 10.2 mg of 1,8-diazabicyclo[5.4.0]-7-undecene (79.0 μmol) were added into 15.0 mL of toluene. Then, 3.10 g of 2-acryloyloxyethyl isocyanate (22.0 mmol, 2.20 eq. to the number of dimethylglyoxime) was added portion-wise into the reaction solution and stirred for 12 h at 60 °C. After cooling to room temperature, the precipitate was washed with 200 mL of hexane. Finally, the precipitate was filtrated and dried under reduced pressure for 5 h at room temperature to yield a white powder (3.82 g, 94% yield). ^1^H-NMR (400 MHz, CDCl_3_, *δ* [ppm]): 6.50 (t, *J* = 5.6 Hz, 2H), 6.39 (dd, *J* = 17.4, 1.2 Hz, 2H), 6.08 (dd, *J* = 17.2, 10.4 Hz, 2H), 5.84 (dd, *J* = 10.4, 1.2 Hz, 2H), 4.27 (t, *J* = 5.2 Hz, 4H), 3.59 (q, *J* = 5.6 Hz, 4H), 2.25 (s, 6H). ^13^C-NMR (100 MHz, CDCl_3_, *δ* [ppm]): 166.61, 158.15, 154.62, 132.08, 128.10, 63.32, 41.20, 12.01.

### 3.3. Preparation of PBA-0.5

To a 10 mL vial bottle, 39.8 mg of DOUDA (0.1 mmol), 9.20 mg of HMPP (56.0 μmol), and 2.565 g of BA (20.00 mmol) were added. The mixture was heated to 110 °C to ensure the dissolution of DOUDA into BA by heating above the melting point of DOUDA (104.3 °C). The acrylate solution was transferred to a handmade polytetrafluoroethylene dish preheated to 110 °C, and subsequently irradiated with UV light of 365 nm with a light intensity of 35 mW/cm^2^ for 600 s at the same temperature to produce PBA-0.5. In this study, eight kinds of cross-linked films were produced by changing the combination of acrylate monomer and the amount of DOUDA, as compiled in Table 2.

### 3.4. General Spectroscopic Measurements

Nuclear magnetic resonance (NMR) spectra were recorded on an NMR spectrometer (JEOL ECZ 400, JEOL, Tokyo, Japan). Deuterated chloroform (CDCl_3_) with tetramethylsilane as an internal standard was used as the solvent in all samples. Chemical shifts are given in parts per million (ppm). Fourier transform infrared (FT-IR) spectra were acquired using an FT-IR spectrometer (FTIR-4700 spectrometer, JASCO, Tokyo, Japan) equipped with an attenuated total reflection (ATR) unit with a diamond prism (ATR Pro One, JASCO).

### 3.5. Swelling Tests

Swelling tests were performed by immersing cut pieces of cross-linked films (10–20 mg) in 2.0 mL of tetrahydrofuran for 24 h at room temperature. The swelling ratio was calculated using Equation (4). In this study, *m*_i_ and *m*_s_ mean the weight of the sample at the initial and swelling state, respectively. The tests were performed for three samples cut at different positions from the same cross-linked polymer films. The swelling ratio was given as the average of these experimental values.
(4)$$Swelling\;ratio(\%)=\frac{m_{s}-m_{i}}{m_{i}}\times 100$$

### 3.6. Rheological and Stress Relaxation Measurements

The storage (*G*’) and loss (*G*”) moduli were assessed using a stress-controlled rheometer (MCR 102, Anton Paar, Graz, Austria) equipped with an 8 mm diameter stainless-steel parallel plate (PP08, Anton Paar, Graz, Austria) and a forced convection oven for temperature control (CTD450, Anton Paar, Graz, Austria). The measurements were performed at 30 °C, at a strain amplitude of 0.3%, and in an angular frequency (*ω*) range between 10^−1^ and 10^2^ rad/s.

The temperature dependence of *G*’ and *G*” were also measured using the same rheometer equipped with a temperature controller from a liquid nitrogen source (EVU20, Anton Paar, Graz, Austria). The measurements were performed at 6.28 rad/s while heating at a rate of 3 °C/min. The measurements were started at ~10 °C below the glass transition temperature (*T*_g_) of each acrylate homopolymer. During the measurements, the strain amplitude was adjusted to 0.01–0.1% or 1.1%, corresponding to the linear regime. The normal force of 0.5–7.0 N was applied to avoid slipping the cross-linked film. In this study, the *T*_g_ was defined as the temperature at which the *G*” value becomes maximum.

Stress relaxation measurements were conducted using the same rheometer. The disk-shaped sample of 8 mm in diameter was prepared using a cutting die. The measurements were performed at the temperature between 150 °C and 180 °C by applying ~3% strain which was within the linear regime.

### 3.7. Compression Tests

Mechanically compression tests were conducted by a compact tabletop testing apparatus (EZ-LX, Shimadzu, Kyoto, Japan). The 4 mm diameter-disk shape samples were prepared using a cutting die and compressed at 0.5 mm/min.

### 3.8. Reprocessing Tests

To confirm reprocessability, POA-0.5 was broken into pieces, inserted between glass substrates, and hot-pressed under 200 °C for 10 min to form the film into a single sheet again. After hot-pressing, the sheet was subjected to stress relaxation measurement, FT-IR spectrum measurement, rheology measurement, and a compression test using the same procedures mentioned above. These measurements were repeated for the sheet after forming it into a single sheet by hot-pressing.

## 4. Conclusions

In this study, we investigated the rheological relaxation properties of a series of CANs from polyacrylates cross-linked via oxime–urethane bonds as dynamic covalent bonds. Such CANs were fabricated by changing the combination of an acrylate monomer and the amount of a diacrylate cross-linker with oxime–urethane bonds. The rheological relaxation related to the oxime–urethane bond exchange reaction was dominant only when fabricated with normal alkyl acrylates such as butyl acrylate and octyl acrylate. Moreover, it was found that bond exchange reactions are greatly hindered by aggregated cross-linkers when fabricated utilizing acrylate monomers containing oxygen atoms arising from the low miscibility of the cross-linker since the relaxation time becomes ~15 times greater when compared to CANs utilizing normal alkyl acrylates. We believe that this research provides a promising guideline for designing and fabricating new polyacrylate-based CANs to contribute toward a sustainable society.

## Figures and Tables

**Figure 1 ijms-25-12897-f001:**
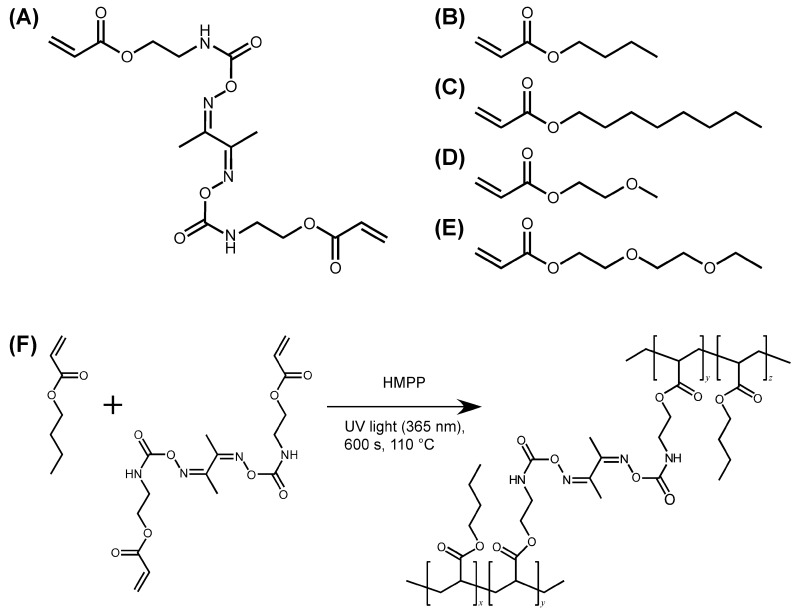
Chemical structures of reagents used in this study and the reaction scheme of cross-linked film. (**A**) Diacrylate cross-linker for possessing oxime–urethane bonds (DOUDA) at the central part. (**B**–**E**) Butyl acrylate (BA), octyl acrylate (OA), 2-methoxyethyl acrylate (MEA), and 2-(2-ethoxyethoxy)ethyl acrylate (EEA) as monomers for the fabrication of the cross-linked films. (**F**) Reaction scheme of cross-linked film using BA as a monomer. *x*, *y* and *z* are the degree of polymerization of monomer and cross-linker.

**Figure 2 ijms-25-12897-f002:**
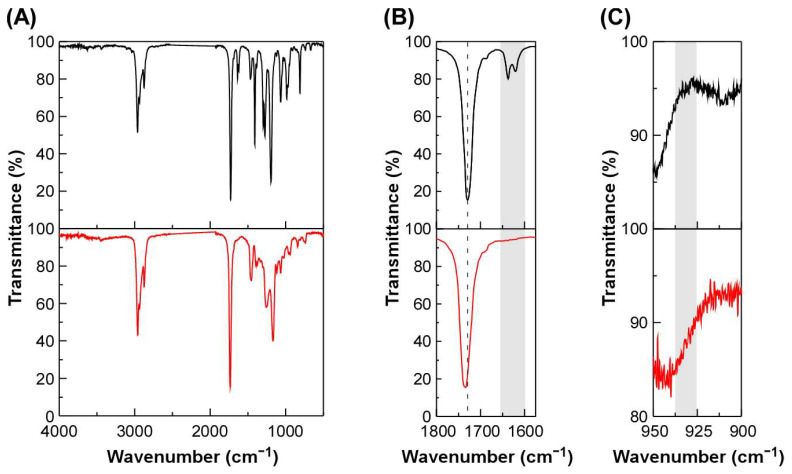
Attenuated total reflection (ATR) FT-IR spectra of the BA (black line) and PBA-0.5 of the UV-irradiated side (red line). (**A**) ATR FT-IR spectra in the wavenumber range between 500 cm^−1^ and 4000 cm^−1^. Magnified FT-IR spectra in the wavenumber range between 1575 cm^−1^ and 1800 cm^−1^, (**B**) and between 900 cm^−1^ and 950 cm^−1^; (**C**) are also shown for reader’s clarity.

**Figure 3 ijms-25-12897-f003:**
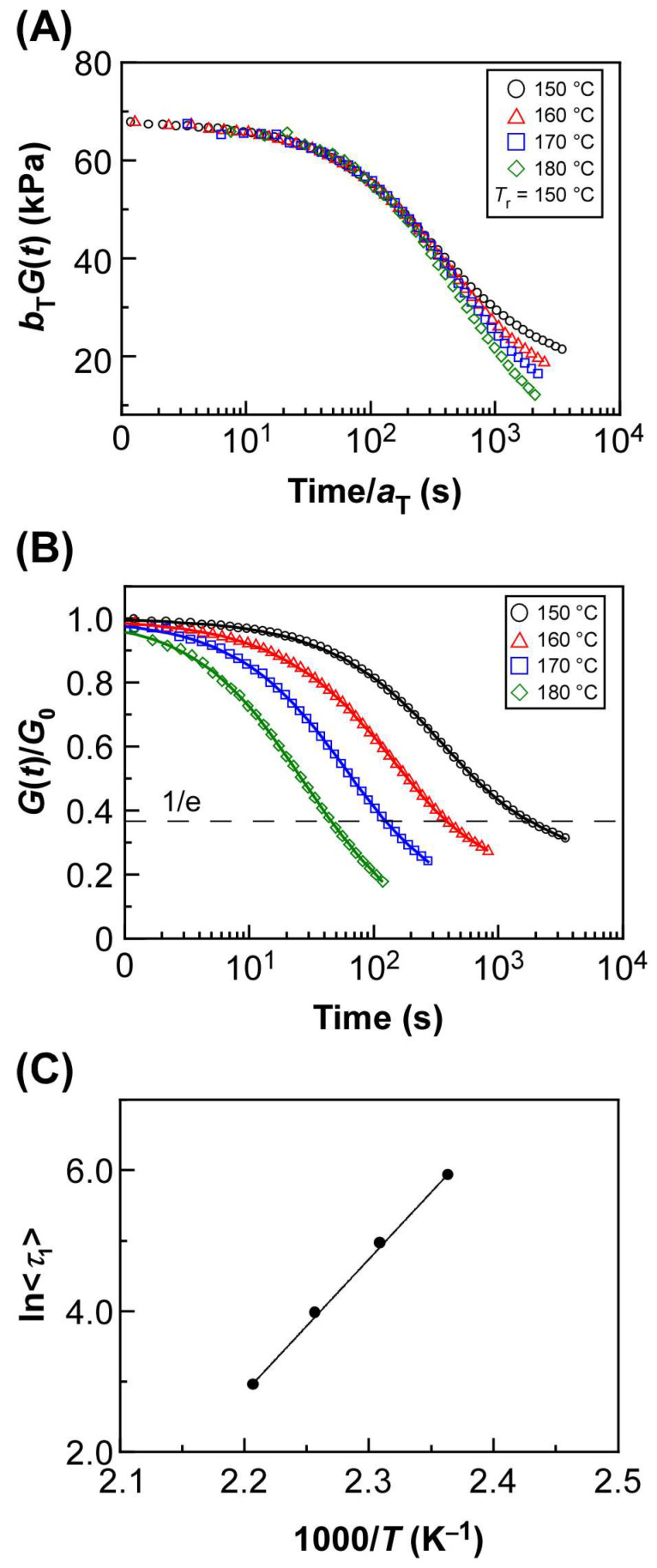
Results of stress relaxation measurements of PBA-0.5. (**A**) Master curve constructed at a reference temperature of 150 °C. The values of *a*_T_ and *b*_T_ are horizontal and vertical shift factors, respectively. (**B**) Stress relaxation curves measured between 150 °C and 180 °C. Solid lines represent the theoretical fitting curves based on Equation (1). (**C**) Arrhenius plot of <*τ*_1_>. Solid line represents the linear regression line of the plots.

**Figure 4 ijms-25-12897-f004:**
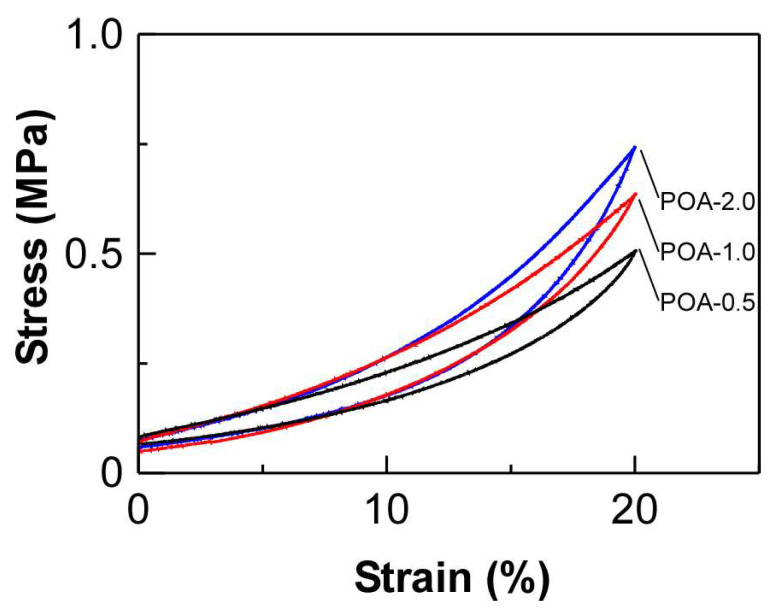
Stress–strain curves of POA-0.5 (black line), POA-1.0 (red line), and POA-2.0 (blue line), compressing the strain up to 20%.

**Figure 5 ijms-25-12897-f005:**
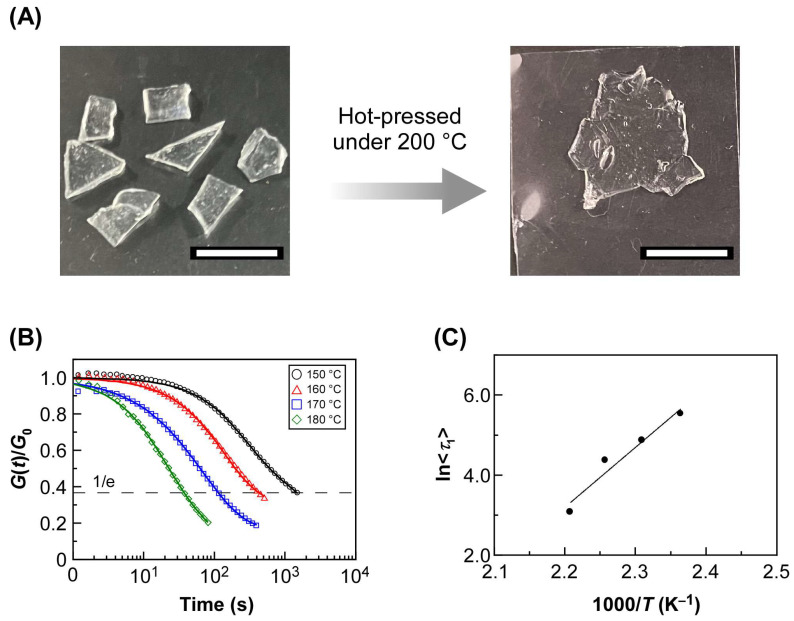
Results of reprocessing test of POA-0.5. (**A**) Photographs of pieces of broken POA-0.5 (left side) and their forming into a single sheet a single sheet by heat pressing at 200 °C for 10 min (right side). White scale bars in the photographs represent 10 mm. (**B**) Stress relaxation curves measured between 150 °C and 180 °C. Solid lines represent the fitting curves based on Equation (1). (**C**) Arrhenius plots of <*τ*_1_>. The solid line represents the linear regression line of the plots.

**Table 1 ijms-25-12897-t001:** Overview of fabrication conditions and physical properties of the cross-linked films.

Sample Code	Acrylate Monomer	Molar Ratio of Monomer and DOUDA	Swelling Ratio (%) *^a^*	*T*_g_ (°C) *^b^*	*G*’ (kPa) *^c^*	*E*_a,sf_ (kJ/mol) ^*d*^	*E*_a,1_ (kJ/mol) ^*e*^	*E*_a,2_ (kJ/mol) ^*f*^
PBA-0.5	BA	200:1	745	−46.4	20.0	153	158	273
PBA-1.0	BA	200:2	477	−45.1	55.5	138	128	260
PBA-2.0	BA	200:4	304	−44.3	180	133	136	199
POA-0.5	OA	200:1	909	−78.4	14.4	144	138	104
POA-1.0	OA	200:2	601	−81.0	33.4	138	132	165
POA-2.0	OA	200:4	428	−80.2	63.3	141	139	197
PMEA-0.5	MEA	200:1	515	−32.1	265	-	-	-
PEEA-0.5	EEA	200:1	591	−56.3	42.2	-	-	-

*^a^* Swelling ratio in tetrahydrofuran (THF) calculated using Equation (4). *^b^* Glass transition temperature (*T*_g_) determined by rheological measurement, which was defined as the temperature at which the loss modulus (*G*”) reaches its maximum. *^c^* Storage modulus (*G*’) at 1.0 Hz measured at 30 °C. *^d^* Activation energy (*E*_a,sf_) calculated from the horizontal shift factor (*a*_T_) of the master curve. *^e^* Activation energy (*E*_a,1_) of the bond exchange reaction. *^f^* Activation energy (*E*_a,2_) of relaxation arising from the melting of aggregated DOUDA.

**Table 2 ijms-25-12897-t002:** Fabricating conditions of the cross-linked films with DOUDA cross-linker.

Sample Code	Acrylate Monomer	Weight of Monomer (g)	Weight of DOUDA (mg)	Weight of HMPP (mg)
PBA-0.5	BA	2.565	39.8	9.2
PBA-1.0	BA	2.565	79.8	13.1
PBA-2.0	BA	2.565	159.3	11.2
POA-0.5	OA	3.696	39.9	9.7
POA-1.0	OA	3.696	79.4	10.6
POA-2.0	OA	3.696	158.8	11.9
PMEA-0.5	MEA	2.606	39.7	10.2
PEEA-0.5	EEA	3.764	39.8	10.5

## Data Availability

Data is contained within the article.

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
