# Peer review of "Covalent Adaptable Networks from Polyacrylates Based on Oxime–Urethane Bond Exchange Reaction"

_ijms, 2024, doi:10.3390/ijms252312897_

Round 1
Reviewer 1 Report
Comments and Suggestions for Authors
The manuscript "Covalent Adaptable Networks from Polyacrylates Based on Oxime-Urethane Bond Exchange Reaction" reports rheological relaxation behavior of covalent adaptable networks based on acrylates incorporating oxime-urthane bonds. The authors analyzed the miscibility between monomers and crosllinking agents and they found that the method is reproducible.
Abstract: the abbreviation CANs at line 19 must be moved after the first mention of Covalent adaptable networks.
Introduction: The motivation of choosing polyurethanes is missing. Which are the advantages over other dynamic bonds? What reports are already available in literature? What properties are for interest? Why the rheological relaxations are important for self-healing ability?
The novelty of this study is not emphasized! The authors used some already published working protocols as they mentioned in section 2.
lines 68-70: there are some repeated information, the phrase must be rewritten!
Section 2.1. The reaction scheme must be presented. The formation of new bonds must be highlighted in IR spectra. The spectra were not evaluated according to the structure! The presence of oxime-polyurethane bonds must be emphasized!The differences must be discussed according to other literature reports!
Fig. 2. Values on Y axis are missing!
line 120: why the molar ratio of DOUDA was very low?
Fig. 4. What about at repeated cycles? How was monitored the different flexibility of the films?
Section 2.3. The thermal stability of films must be evaluated. Some oxidation processes are mentioned! Is the initial structure preserved? This must be deeply investigated.
Section 3.5. is not discussed! The results are missing!
I recommend this paper to be major revised before the acceptance!
Author Response
Manuscript ID: ijms-3314370
Type of manuscript: Article
Title: Covalent Adaptable Networks from Polyacrylates Based on Oxime-Urethane Bond Exchange Reaction
Authors: Yu Sotoyama, Naoto Iwata, Seiichi Furumi*
Point-by-point responses. Our responses and opinions are shown in BLUE.
Reviewer 1
To Reviewer 1
Thank you very much for your kind review and generous comments on our report. We have prepared the revised manuscript according to your comments. We are looking forward to hearing your honest opinions on our revised manuscript.
Comment #1
Abstract:
The abbreviation CANs at line 19 must be moved after the first mention of Covalent adaptable networks.
Thank you for pointing out our mistake. The abbreviation of CAN was moved after the first mention of Covalent adaptable networks (Line 8, Page 1 in this revised manuscript).
Comment #2-a
Introduction:
The motivation of choosing polyurethanes is missing. Which are the advantages over other dynamic bonds?
Transesterification is one of the frequently adopted bond exchange reactions for the fabrication of CANs. However, this is not suitable for creating acrylate-based CANs because ester moieties of polyacrylates may also contribute to the bond exchange reaction. Additionally, transesterification requires an additional base catalyst, which must have good miscibility to the polymer network (Ref. No. 25; Hayashi, M. et al., Preparation of Colorless, Highly Transparent, Epoxy-Based Vitrimers by the Thiol-Epoxy Click Reaction and Evaluation of Their Shape-Memory Properties. ACS Applied Polymer Materials, 2020, 2, 2452–2457.). On the other hand, oxime-urethane bond exchange reaction proceeds in the absence of a catalyst while preserving ester moieties of polyacrylates. According to your suggestion, we revised the sentences in Introduction (Lines 63–73, Page 2 in this revised manuscript) as follows:
In this study, we adopted oxime-urethane bonds as a kind of dynamic covalent bond because the oxime-urethane bond exchange reaction progresses independently regardless of the presence of ester bonds in polyacrylates. In previous studies, polyurethane-based thermoset materials and elastomers incorporating oxime-urethane bonds as dynamic covalent bonds have been fabricated [19–24]. Moreover, oxime-urethane bond exchange reaction proceeds even in the absence of a catalyst while preserving ester moieties of polyacrylates. This is very beneficial from the experimental perspective because CANs utilizing transesterification reaction require a base catalyst, which must have good miscibility to the polymer network [25]. By copolymerization of acrylate monomers with a diacrylate cross-linker with an oxime-urethane bond (DOUDA), a new class of CANs based on polyacrylates were successfully prepared.
Comment #2-b
Introduction:
What reports are already available in literature?
In previous literature, polyurethane-based thermoset materials and elastomers incorporating oxime-urethane bonds as dynamic covalent bonds have been fabricated (Ref. Nos. 19–24).
Comment #2-c
Introduction:
What properties are for interest?
We found it fascinating that, unlike transesterification, oxime-urethane bond exchange reaction proceeds without additional catalyst. It is also important to note that fabrication of polyacrylate-based CANs has not been widely investigated.
Comment #2-d
Introduction:
Why the rheological relaxations are important for self-healing ability?
In the case of conventional cross-linked polymers, the applied stress does not relax completely due to the formation of a three-dimensional polymer network structure by static bonds. On the other hand, stress relaxation is observed for CANs due to the activation of bond exchange reaction. Relaxation time can be used to evaluate the mobility of CANs when heated. Stress relaxation measurements are often used to evaluate the self-healing properties of CANs, and were also used for associative CANs since the first report by Leibler and colleagues (Ref. No. 11; Leibler, L. et al., Silica-Like Malleable Materials from Permanent Organic Networks, Science, 2011, 334, 965–968).
Comment #3
The novelty of this study is not emphasized! The authors used some already published working protocols as they mentioned in section 2.
The novelty of this study is that we have revealed that the miscibility of the cross-linker and monomer has a significant impact on the self-healing properties of the polyacrylate-based CANs. We adopted previous methods for the synthesis of cross-linkers and the evaluation of cross-linked films for the fair comparison with precedents by You and colleagues (Ref. No. 26; You, Z. et al., Simple Solvent-Free Strategy for Synthesizing Covalent Adaptable Networks from Commodity Vinyl Monomers, Macromolecules, 2021, 54, 4081–4088,).
Comment #4
lines 68-70: there are some repeated information, the phrase must be rewritten!
Based on your suggestion, we have revised the sentences (Lines 73–77, Page 2 in this revised manuscript) as follows:
Moreover, we investigated the reprocessability of a series of cross-linked polyacrylate films fabricated by changing the combination of acrylate monomer and the amount of cross-linker. It was indicated that the miscibility of DOUDA is very important for the fabrication of CANs with excellent reprocessability because aggregated DOUDA hindered the bond exchange reaction.
Comment #5-a
Section 2.1. The reaction scheme must be presented.
Based on your suggestion, we have added newly the reaction scheme in Figure 1(F).
Comment #5-b
The formation of new bonds must be highlighted in IR spectra. The spectra were not evaluated according to the structure! The presence of oxime-polyurethane bonds must be emphasized! The differences must be discussed according to other literature reports!
According to your suggestion, we confirmed a decrease in transmittance originating from N-O bond was observed ~927 cm−1, indicating that the oxime-urethane bonds are incorporated into the cross-linked film. To explain this result, we have added following sentence. (Lines 96–98, Pages 2 and 3 in this revised manuscript):
Furthermore, a decrease in transmittance originating from N-O bond was observed at ~927 cm−1 after copolymerization with DOUDA (Figure 2).
Comment #6
Fig. 2. Values on Y axis are missing!
Thank you for pointing out our mistake. Values on Y axis have been added. In addition, we confirmed that the cross-linked film contains N-O bond due to DOUDA inside the cross-linked film.
Comment #7
line 120: why the molar ratio of DOUDA was very low?
When the molar ratio of monomers and the cross-linker up to 200:8, the cross-linked film became whitely turbid. This result indicated that the aggregated cross-linkers caused light scattering. To explain these results, we have added transmission spectra and photographs of POA-4.0 and POA-8.0 whose monomer to cross-linker molar ratios are 200:8 and 200:16, respectively (Figure S11). Additionally, the following sentences have been added in Lines 233–239, Page 8 in this revised manuscript.
However, it should be noted that the mole fraction of DOUDA must be less than 200:4, even for monomers with relatively good miscibility such as BA and OA. In fact, when POA-4.0 and POA-8.0, that is, cross-linked films with a 200:8 or 200:16 molar fraction of DOUDA were prepared, both films were whitely turbid. Although the transmittance of POA-0.5 was 90%, that of POA-4.0 and POA-8.0 was less than 10% (Figure S11). Such significant decrease of transmitted light strongly suggests that the aggregated DOUDA causes the light scattering phenomenon.
Comment #8
Fig. 4. What about at repeated cycles? How was monitored the different flexibility of the films?
After repeated cycles of compression and release, the maximum point stress gradually decreased. The flexibility of the film was evaluated by the maximum point stress and the storage modulus at 1 Hz at 30 ºC. In addition, we commented about Young's modulus and hysteresis loss rate of the films in Section 2.2 (Lines 254–261, Page 9 in this revised manuscript) as follows:
Figure 4 shows the stress-strain curves of POA-0.5, POA1.0, and POA-2.0 in compression process with the stress up to 20%. The Young's modulus was ~13 kPa regardless of the difference in the molar ratio of DOUDA. The hysteresis loss rate was ~24% for all films, which is comparable to that of commercially-available chloroprene rubbers. This suggested that bond exchange reaction is completely frozen at room temperature similar to the conventional thermosets. The Young's modulus remained constant upon repeated circles of compression, while the maximum point stress gradually decreased to 84% of the initial value (Table S9 and Figure S14).
Comment #9
Section 2.3. The thermal stability of films must be evaluated. Some oxidation processes are mentioned! Is the initial structure preserved? This must be deeply investigated.
After reprocessing, the storage and loss moduli were slightly reduced compared to those of as prepared. This experimental fact suggests that POA-0.5 has not significantly decomposed. We are also in the process of performing thermogravimetric measurements to confirm that POA-0.5 is not significantly decomposed at 200 ºC.
Comment #10
Section 3.5. is not discussed! The results are missing!
The results of swelling test have been already discussed in section 2.1.(Lines 99–102, Page 2 in this revised manuscript). All data of swelling test added to Table1.

Reviewer 2 Report
Comments and Suggestions for Authors
The paper is devoted for covalent adaptable networks from polyacrylates based on oxime-urethane bond exchange reaction. The topic is generally interesting, however the paper contain unexplained places (below) and need major revisions.
Fig. 2, please explain why infrared investigations were performed only in frequency range 4000 – 500 cm-1?
Line 19, please explain the term ‘’excellent relaxation properties’’.
Figure 4 should be more commented.
Conclusions should be rewritten in more informative way.
Abbreviations should be explained by first using, for example line 19 CAN.
Author Response
Manuscript ID: ijms-3314370
Type of manuscript: Article
Title: Covalent Adaptable Networks from Polyacrylates Based on Oxime-Urethane Bond Exchange Reaction
Authors: Yu Sotoyama, Naoto Iwata, Seiichi Furumi*
Point-by-point responses. Our responses and opinions are shown in BLUE.
Reviewer 2
To Reviewer 2
Thank you very much for your kind review and generous comments on our manuscript. We have prepared the revised manuscript according to your comments. We are looking forward to hearing your honest opinions on our revised manuscript.
Comment #1
Fig. 2, please explain why infrared investigations were performed only in frequency range 4000 – 500 cm-1?
The FT-IR spectra are generally measured in the range from 500–4000 cm−1. In the Sigma-Aldrich Library of FT-IR Spectra, the spectra are measured in the range of 400–4000 cm−1.
Comment #2
Line 19, please explain the term ‘’excellent relaxation properties’’.
Ideally, bond exchange reaction of CANs should be completely freeze at operating temperatures as they exhibit mechanical strength that is comparable to conventional thermosets. In contrast, bond exchange reaction of CANs should be faster upon elevated temperatures because heating at high temperatures for long a long time may induce the degradation of CANs. From this reason, we consider that CANs with high temperature dependence of relaxation time, that is, with higher Ea have excellent relaxation properties. We rewrote the sentence (Lines 18–20, Page 1 in this revised manuscript) to make it easier to understand as follows:
Moreover, it was also indicated that the miscibility of the cross-linker is very important for the fabrication of CANs with good self-healing ability and reprocessability.
Comment #3
Figure 4 should be more commented.
According to your comment, we commented about Young's modulus and hysteresis loss rate of the films in Section 2.2 (Lines 254–261, Page 9 in this revised manuscript) as follows:
Figure 4 shows the stress-strain curves of POA-0.5, POA1.0, and POA-2.0 upon 20% compression. The Young's modulus was ~13 kPa regardless of the difference in the molar ratio of DOUDA. The hysteresis loss rate was ~24% for all films, which is comparable to that of commercially-available chloroprene rubbers. This suggested that bond exchange reaction is completely frozen at room temperature similar to the conventional thermosets. The Young's modulus remained constant upon repeated circles of compression, while the maximum point stress gradually decreased to 84% of the initial value (Table S9 and Figure S14).
Comment #4
Conclusions should be rewritten in more informative way.
Based on your suggestion, we have rewritten conclusion (Lines 369–381, Page 12 in this revised manuscript) using specific numerical values and compound names as follows:
In this report, we investigated the rheological relaxation properties of a series of CANs from polyacrylates cross-linked via oxime-urethane bonds as dynamic covalent bonds. Such CANs were fabricated by changing the combination of acrylate monomer and the amount of a diacrylate cross-linker with oxime-urethane bonds. The rheological relaxation related to the oxime-urethane bond exchange reaction was dominant only when fabricated with normal-alkyl acrylates such as butyl acrylate and octyl acrylate. Moreover, it was found that bond exchange reaction is greatly hindered by aggregated cross-linkers when fabricated utilizing acrylate monomers containing oxygen atoms arising from the low miscibility of the cross-linker because the relaxation time became ~15 times larger when compared to CANs utilizing normal-alkyl acrylates. We believe that this report provides a promising guideline to design and fabricate new polyacrylate-based CANs for the contribution to a sustainable society.
Comment #5
Abbreviations should be explained by first using, for example line 19 CAN.
Thank you for pointing out our mistake. CAN was moved after the first mention of Covalent adaptable networks (Line 8, Page 1 in this revised manuscript).

Reviewer 3 Report
Comments and Suggestions for Authors
This paper describes the rheological relaxation properties of a series of covalent adaptable networks (CANs) from polyacrylates cross-linked via oxime-urethane bonds as dynamic covalent bonds. The authors fabricated a series of cross-linked polyacrylate films by changing the combination of acrylate monomer and the amount of diacrylate cross-linker possessing oxime-urethane bonds to evaluate their rheological relaxation properties and found that the contribution from the relaxation due to the bond exchange reaction becomes dominant only when the normal-alkyl acrylates are used as a monomer. Moreover, they indicated that the miscibility of the cross-linker is very important for the fabrication of CANs with excellent stress relaxation property to exhibit self-healing ability. I think the experiments were carefully done and the results are reliable. I would like to accept this manuscript in International Journal of Molecular Sciences. This paper will give useful information in the field of cross-linked polymers composed of covalent adaptable networks. The only thing I am curious is only a small amount of added cross-linking agent (DOUDA) can aggregate.
Author Response
Manuscript ID: ijms-3314370
Type of manuscript: Article
Title: Covalent Adaptable Networks from Polyacrylates Based on Oxime-Urethane Bond Exchange Reaction
Authors: Yu Sotoyama, Naoto Iwata, Seiichi Furumi*
Point-by-point responses. Our responses and opinions are shown in BLUE.
Reviewer 3
To Reviewer 3
Thank you very much for your kind review and generous comments on our report. We have prepared the revised manuscript according to your comments. We are looking forward to hearing your honest opinions on our revised manuscript.
Comment #1
The only thing I am curious is only a small amount of added cross-linking agent (DOUDA) can aggregate.
The fact that the cross-linker did not dissolve when mixed at room temperature with all monomers suggests that the miscibility between the cross-linker and monomers is quite low. In fact, increasing the molar ratio of monomers and the cross-linker up to 200:8 caused the cross-linked film to scattered light. However, some monomers could be used as CANs because the miscibility varied with the type of monomer. We added transmission spectra and photographs of POA-4.0 and POA-8.0 whose monomer to cross-linker molar ratios are 200:8 and 200:16, respectively (Figure S11) and mentioned it in the main text of revised manuscript (Lines 233–239, Page 8 in this revised manuscript) as follows:
However, it should be noted that the mole fraction of DOUDA must be less than 200:4, even for monomers with relatively good miscibility such as BA and OA. In fact, when POA-4.0 and POA-8.0, that is, cross-linked films with a 200:8 or 200:16 molar fraction of DOUDA were prepared, both films were whitely turbid. Although the transmittance of POA-0.5 was 90%, that of POA-4.0 and POA-8.0 was less than 10% (Figure S11). Such significant decrease of transmitted light strongly suggests that the aggregated DOUDA causes the light scattering phenomenon.

Round 2
Reviewer 1 Report
Comments and Suggestions for Authors
The authors provided sufficient data to support their research and the paper was improved. The manuscript can be accepted for publication in the revised form.
Reviewer 2 Report
Comments and Suggestions for Authors
Authors make proper corrections according to reviewer remarks and I suggest publish the paper as it is.